# Cellular Senescence in the Lung: The Central Role of Senescent Epithelial Cells

**DOI:** 10.3390/ijms21093279

**Published:** 2020-05-06

**Authors:** Christine Hansel, Verena Jendrossek, Diana Klein

**Affiliations:** Institute of Cell Biology (Cancer Research), University of Duisburg-Essen, University Hospital Essen, Virchowstrasse 173, 45122 Essen, Germany; Christine.Hansel@uk-essen.de (C.H.); verena.jendrossek@uk-essen.de (V.J.)

**Keywords:** senescence-associated secretory phenotype, SASP, lung injury, pulmonary disease, radiotherapy, ionizing radiation, cancer therapy

## Abstract

Cellular senescence is a key process in physiological dysfunction developing upon aging or following diverse stressors including ionizing radiation. It describes the state of a permanent cell cycle arrest, in which proliferating cells become resistant to growth-stimulating factors. Senescent cells differ from quiescent cells, which can re-enter the cell cycle and from finally differentiated cells: morphological and metabolic changes, restructuring of chromatin, changes in gene expressions and the appropriation of an inflammation-promoting phenotype, called the senescence-associated secretory phenotype (SASP), characterize cellular senescence. The biological role of senescence is complex, since both protective and harmful effects have been described for senescent cells. While initially described as a mechanism to avoid malignant transformation of damaged cells, senescence can even contribute to many age-related diseases, including cancer, tissue degeneration, and inflammatory diseases, particularly when senescent cells persist in damaged tissues. Due to overwhelming evidence about the important contribution of cellular senescence to the pathogenesis of different lung diseases, specific targeting of senescent cells or of pathology-promoting SASP factors has been suggested as a potential therapeutic approach. In this review, we summarize recent advances regarding the role of cellular (fibroblastic, endothelial, and epithelial) senescence in lung pathologies, with a focus on radiation-induced senescence. Among the different cells here, a central role of epithelial senescence is suggested.

## 1. Introduction

Cellular senescence and in particular the ‘cellular senescence phenotype’ was initially discovered by Leonard Hayflick in 1961, who observed that the number of cell divisions in fibroblasts was limited and these normal, non-transformed cells reached the end of their replicative life span upon prolonged culturing, because the telomeres had reached a critical length [1,2]. In contrast to embryonic cells and stem cells, normal human cells divide approximately 52 times (the so-called Hayflick limit) before cell aging finally begins by entering a state of permanent growth arrest. Herein, the gradual shortening of telomeres (30–200 bp with each mitotic event) is thought to manifest an increased incidence of double-strand breaks (DSBs) at the DNA ends (known as ‘the end replication problem’) [3,4]. DSB (and even DNA single-strand breaks) in turn trigger the activation of DNA damage response (DDR) pathways, finally leading to the activation of the downstream kinases CHK2 and CHK1 through the protein kinases Ataxia-telangiectasia-mutated (ATM) and Ataxia-telangiectasia and Rad3-related (ATR), respectively [5]. The resulting cell cycle arrest allows for effective DNA repair, and thus for the resumption of normal cell functioning [5]. Increased DNA damage and/or inefficient damage removal then results in chronic DDR signaling that can foster apoptotic cell death or a stable cell cycle arrest—cellular senescence [5,6]. This senescence-associated cell cycle arrest (mostly G1) depends on the activation of the cyclin-dependent kinase (CDK) inhibitors p21/WAF1 and p16/INK4A, the decisive components of tumor-suppressor pathways that are governed by the p53 and retinoblastoma (Rb) proteins, respectively [7,8].

The cellular senescence phenotype bears morphological as well as characteristic gene expression alterations. Senescent cells have an enlarged, flattened and irregular shape bearing more vacuoles, an increase in senescence-associated β-galactosidase (SA-β-gal) activity (due to more and bigger lysosomes), and (partially) an altered chromatin organization known as senescence-associated heterochromatin foci (SAHF) [9]. The stable and irreversible form of cell cycle arrest is due to accumulation of the p16, p15/INK4b, p27/Kip1 and p21 CDK inhibitors, while the cell’s metabolic activity is maintained. Furthermore, tumor suppressor proteins, such as phosphatase PTEN (phosphatase tensin homolog), p53 or hypo-phosphorylated Rb, can be used to detect cellular senescence. Even the absence of markers can be used, including the absence of the proliferation marker Ki-67 or the lack of bromodeoxyuridine (BrdU) incorporation. Thus, there are reliable methods to specifically detect senescent cells, but due to the lack of a ‘master senescent cell marker’, usually a combination of markers is necessary to detect cellular senescence in vitro and in vivo in isolated tissues of interest (Table 1). Of note, most of these tools lack the capability of real-time imaging of senescence particularly in living subjects [10]. This is a critical issue with respect to practical applications such as image-guided surgical removal of senescent cells, as well as monitoring senescence during different pathologies [11,12,13]. Non-invasive biomarkers for cellular senescence could include the senescence-associated secretory phenotype (SASP), that involves the production of secretory growth factors and cytokines, reinforce the senescence arrest, and alter the cell’s microenvironment, e.g., changes in extracellular matrix composition and the immune environment [9,14,15,16]. Interleukin-6 (IL-6) and IL-8 for example are key SASP factors boosting the senescence growth arrest by acting in an autocrine feedback loop. Further examples of well-known SASP factors are: the CC-chemokine ligand 2 (CCL2), a monocyte chemotactic protein with even angiogenic potential, as well as factors like transforming growth factor beta (TGFβ), a multifunctional and pro-fibrotic cytokine, plasminogen activator inhibitor-1 (PAI-1), also known as endothelial plasminogen activator inhibitor (or serpin E), and insulin-like growth factor 1 (IGF-1), which plays an important role in childhood growth, and has anabolic effects in adults [17,18,19,20,21]. Upon secretion from senescent cells, these SASP factors usually act in a paracrine manner to stimulate proliferation and/or transformation of adjacent immortalized cells, or even might trigger the senescence of other cells in the microenvironment.

It is thought that cellular senescence contributes to developmental processes including promoting remodeling, inflammation, infectious susceptibility, and angiogenesis as well as fundamental processes, such as wound healing and tissue regeneration. Herein, senescent cells which fulfilled their action are removed from the interfered tissue via infiltrating immune cells. However, if senescent cells persist, these cells might foster age- and disease-associated physiological dysfunction particularly through their progressively changing secretory profile [46].

With this respect, cellular senescence is now considered an important driving force for the development of chronic lung pathologies, particularly chronic inflammation observed in lungs of aging patients and of patients suffering from asthma, chronic obstructive pulmonary diseaseor pulmonary fibrosis. The accumulation of senescent cells in lungs has disadvantageous consequences [47,48]. Understanding the mechanisms driving induction of cellular senescence as well as the mechanisms mediating pathology-promoting effects of senescence may offer new treatment strategies for chronic lung diseases. In this review, we summarize recent findings about the different senescent lung cells with respect to their potential contribution to inflammation and remodeling/fibrosis, and with a special focus on the contribution to radiation-induced pneumopathy.

## 2. Replicative Senescence Versus Stress-Induced Senescence

In addition to replicative senescence, the type of senescence that is characterized by the shortening of telomeres, another type of senescence, namely stress-induced premature senescence (SIPS), has been described [49]. Increased cellular stress, as mediated by the activation of oncogenes, loss and/or inactivation of tumor suppressor genes, different agents leading to the accumulation of DNA damage such as genotoxic drugs or doses of radiation, the presence of reactive oxygen species (ROS) or metabolic and epigenetic changes can foster cells entering a postmitotic state (2–3 days after stress-exposure), in which the cells then display signs of senescence [50]. Interestingly, replicative senescence and stress-induced senescence share a similar morphological appearance as well as biomarkers used to detect the induction of senescence. For example, cells undergoing stress-induced senescence after sub-cytotoxic stresses develop a characteristic senescent morphology, an irreversible growth arrest (at the G1/S phase of the cell cycle) with overexpression of several CDK inhibitors and hypo-phosphorylation of Rb, as well as the presence of SA-β-gal activity, and thus biomarkers reminiscent of replicative senescence at a long-term [51,52,53]. Even at the molecular level, senescence markers were shared between replicative senescent and stress-induced senescent fibroblasts: fibronectin, SM22, osteonectin, apolipoprotein J, SS9, the α-subunit of the GTP binding proteins and α1(I)-procollagen were found to be overexpressed in senescent cells, regardless of the type of senescence [52,54].

However, stress-induced and replicative senescence differed at protein level; therefore, differentially expressed proteins observed in stress-induced premature senescent fibroblasts were classified as (i) changes common with replicative senescence, (ii) changes specific to each kind of stressor, and (iii) changes related to SIPS independently of the nature of the stress applied [53]. Gene expression analyses of human diploid fibroblasts later on identified candidate genes being characteristic for premature senescent fibroblasts (induced by the chemical stressors tert-butylhydroperoxide or ethanol) as compared to replicative senescence: these included genes involved in growth arrest (PTEN, insulin-like growth factor binding protein-3, low density lipoprotein receptor-related protein 1 and caveolin-1), senescent morphogens (TGFβ-1 and lysyl oxidase-like 2) and iron metabolism (transferrin and ferritin light chain) [55].

Importantly, a comparative reconstruction of molecular cascades specific for replicative and stress-induced senescence in human fibroblasts revealed the following molecular characteristics: a serine/threonine kinase Aurora B-driven cell cycle signaling accompanied with the suppression of anabolic branches of the fatty acid and estrogen metabolism might be specific for replicative senescence [56]. In contrast, Aurora B signaling was deprioritized in stress-induced premature senescent fibroblasts (as achieved by bleomycin exposure), and the synthetic branches of cholesterol metabolism turned out to be upregulated, as well as the proteasome/ubiquitin ligase pathways of protein degradation [56].

## 3. Cellular Senescence in Adult Lungs

### 3.1. Age-Related (Replicative) Senescence in Adult Lungs

Age-related changes in lung morphology include enlargement of small airways and a decreased alveolar surface tension, finally leading to a compliant distensible lung. Furthermore, senescent cells with increased SASP secretion accumulate with age in adult lungs; these cells exert autocrine and paracrine effects resulting in increased inflammation, induced stem cell dysfunction, and/or senescence of neighboring cells [57]. Most importantly, the age-related increase in senescent lung cells, together with ‘immune senescence’, namely the lack of inflammatory cells to respond to SASP, result in an ineffective or slowed clearance of senescent cells, a progressively altered local environment, and subsequently tissue aging or development of age-related diseases [57].

In particular, TERT-mediated telomere dysfunction was shown to impact on senescence of lung (epithelial) cells. Telomerase, a ribonucleoprotein complex that consists of the RNA-dependent polymerases, telomerase reverse transcriptase (TERT; catalytic component) and RNA template (TR), lengthens telomeres in DNA strand by catalyzing the addition of nucleotides in a TTAGGG sequence to the ends of a chromosomal telomeres. These additional repetitive sequences prevent degradation of the chromosomal ends following multiple rounds of replications [58,59]. In terminally differentiated cells, like lung epithelial cells, telomerase is usually silenced, and its reactivation supports immortalization and unlimited growth. Mutant TERT, TR, and/or shortened telomeres were shown to increase the risk for pulmonary fibrosis [60,61,62]. Senescent cells that could become postmitotic were able to exceed the Hayflick limit and to become potentially immortal by a continuous telomerase activity [58,59]. Lengthening of telomeres or the reduction in the proportion of shortened telomeres in turn was shown to improve the fibrosis outcome [63]. For example, reconstitution of TERT in TERT-deficient mice reduced bleomycin-induced and senescence mediated pulmonary inflammation and fibrosis [63]. Corroborating these findings, epithelial cells (alveolar epithelial cells type II, AECII) that were generated from AECII-specific TERT conditional knockout mice, showed increased cellular senescence upon bleomycin treatment, and respective mice were characterized by enhanced lung injury, inflammation, and fibrosis [64]. Thus, preventing pulmonary senescence plays a protective role in lung injury and offers new potential therapeutic target sites to address increased cellular senescence.

### 3.2. Cellular Stress-Induced Senescence in Adult Lungs

Beside this age-related or stress-induced replicative senescence observed in lungs, stress-induced premature senescence independently of telomere shortening can be induced upon exposure of several stressors: oncogene activation, loss and/or activation of tumor suppressors, DNA damage, ROS and mitochondrial dysfunction, and thus especially oxidative stress resulting from treatment with ionizing radiation and chemotherapy [7,65]. Oxidative stress, in turn, might even increase the SASP release by already senescent cells following the aging process [57].

#### 3.2.1. Radiation-Induced Cellular Senescence in Lungs

Radiotherapy is used in more than half of all cancer patients, both for curative and palliative purposes [66]. Although modern and precise radiotherapy techniques substantially improved the delivery of energy (summarized as ionizing radiation) in the form of electromagnetic waves (gamma- or X-rays) or particles (neutrons, beta or alpha) used for cancer eradication, damage healthy cells and can lead to severe early and late complications in the tumor microenvironment with an increased risk of morbidity in patients after radiotherapy (RT) [67]. Radiation-induced lung disease (inflammation and fibrosis) is a major hurdle in the successful treatment of thorax-associated tumors [68,69]. Radiation-induced pulmonary fibrosis affects up to 25% of cancer patients receiving radiotherapy to tumors of the thoracic region [70]. The radiosensitivity of lung tissue is also dose-limiting when the whole body is irradiated prior bone marrow transplantation [71,72]. The mechanism of radiation-induced normal tissue damage, however, is not fully understood; no causal strategy for the prevention or treatment of radiation-induced damage to the lungs is available so far [73,74]. Highly conformal radiation techniques such as stereotactic body radiation therapy (SBRT) or intensity-modulated radiotherapy (IMRT) are suited to minimize the irradiated lung volume. For example, SBRT is applied to patients with early stage inoperable non-small cell lung cancer [75]. Through image-guided precise targeting of very small volumes, relatively high dose-per-fraction sizes were delivered to the tumors. However, persisting adverse effects such as chest wall pain, rib fracture, esophagitis, brachial plexopathy, and in particular pneumonitis and fibrosis were reported [75,76]. In contrast to conventional dose rates (1–4 Gy/min), the so called ‘‘Flash” radiotherapy (>40 Gy/s; Flash-RT) was shown to enhance the differential effect between normal tissue and tumor in lung models [77]. Herein, it was hypothesized that the protective effect of Flash irradiation was related to the high dose rate delivery. Indeed, Flash irradiation was shown to minimize persistent DNA damage, to reduce the inflammatory response and to facilitate radiation recovery [78].

Cellular stress, and in particular DNA-damaging drugs and ionizing radiation, can induce senescence in most lung cell types. Persistence of senescent cells in turn remains a major problem in certain lung diseases as these senescent cells or more precisely their altered secretory profile, the so-called SASP, might foster lung injury. However, the precise role of each senescent cell type within lung (radiation) injury remains elusive because it depends on the pathological trigger (e.g., dose and fractionation for RT) and the temporality of the observation [22]. The following sections address the recent findings concerning cellular senescence in lung fibroblasts, endothelial and epithelial cells and with respect to their potential contribution to lung injury and particularly pulmonary fibrosis, with a special focus on radiation-induced lung injury.

##### Senescence of Lung Fibroblasts

As tissue-resident mesenchymal cells, lung fibroblasts are a heterogeneous group of cells that can be found in the interstitial space. Lung fibroblasts are the main producer of extracellular matrices (ECM), and thus are crucial to building a lung structural framework as well as for maintaining the integrity of the alveolar structure [79]. Subsets of fibroblasts, including myofibroblasts, lipofibroblasts, and matrix fibroblasts have constituted around 10% of all lung cells [80]. By regulating quality and stiffness of the ECM, fibroblasts can direct cell growth by modulating cell-cell and cell-matrix contacts as well as by paracrine mechanisms [81]. Quality and stiffness of the ECM in turn affect the fibroblasts behavior; by proliferating and repairing injured areas fibroblasts contribute to the tissues repair processes [82]. Lung injury can induce the activation and proliferation of fibroblasts. Although the cellular contributions and interactions herein are complex, particularly lung fibrotic disorders are characterized by accumulation of fibroblasts, myofibroblasts and a respective overproduction of ECM. Too much or aberrant activity of these processes results then in dysregulated inflammation as well as scarring and fibrosis, finally leading to chronic respiratory failure [83]. Herein, pulmonary fibrosis represents probably the most aggressive and progressive form of interstitial lung disease (ILD), a group of diffuse parenchymal lung disorders, whereas some forms of fibrosis, such as acute lung injury or cryptogenic organizing pneumonia are at least partially reversible [83]. However, therapeutic options remain limited. In addition to ECM production and secretion, lung fibroblasts and particularly activated fibroblasts and myofibroblasts contribute to repair by generating contractile forces. Transmitted to the surrounding ECM then enables the activation of profibrotic inflammatory and profibrotic cytokines including integrin-bound latent TGFβ [84].

Senescent lung fibroblasts were already described to play a critical role in fibrotic lung diseases, e.g., idiopathic pulmonary fibrosis (IPF). Senescent fibroblasts have been identified in the lungs of patients with IPF and in fibroblast cultures from IPF lungs [46]. However, less is known about what finally drives the senescent phenotype of lung fibroblasts. As a key player, the signal transducer and activator of transcription 3 (STAT3), a STAT protein family transcription factor that is phosphorylated by receptor-associated Janus kinases (JAK) in response to cytokines and growth factors, was suggested. Increased STAT3 signaling was shown to correlate with IPF progression and thus modulation of this signaling pathway was thought to target early-stage senescence and restore normal fibroblast function [46].

Concerning radiation-induced lung injury and particularly with respect to radiation-induced pulmonary fibrosis, there is a lack of in vivo investigations addressing radiation-induced senescence of fibroblast as well as their contribution to pulmonary disease. Although the prevailing hypotheses suggest that radiation-induced lung fibrosis is an epithelial-fibroblastic disorder, RT injures predominately affect pulmonary epithelial and endothelial cells, thereby causing the release of proinflammatory cytokines that in turn activate lung fibroblasts and recruit inflammatory cells to the sites of injury [85]. Conformingly, it was further shown that senescent lung epithelial cells and particularly the epithelial-derived SASP-factors, contributed to the activation of pulmonary fibroblasts (including increased migration and proliferation) [35]. Activated fibroblasts further increased the expression of α-SMA (smooth muscle actin) and collagen-I. Mechanistically, the activation of Wnt/β-catenin signaling by senescent epithelial cells fostered aberrant expression of the transcriptional factor Nanog in pulmonary fibroblasts, and Nanog silencing in turn suppressed fibroblast activation and impaired the development of pulmonary fibrosis [35]. Thus, the activation of fibroblasts/ myofibroblasts in response to epithelial/endothelial damage foster then disease progression by complex cell–cell and cell–matrix interactions and by synthesizing and depositing ECM proteins (Figure 1).

Various in vitro investigations, however, suggest that therapeutic irradiation might directly affect fibroblasts by inducing growth arrest and senescence. For example, human lung fibroblasts exposed to repeated sub-cytotoxic doses of γ-irradiation (with 4 Gy up to a cumulative dose of about 50 Gy) underwent p53-dependent cellular senescence and these prematurely senescent fibroblasts enhanced the growth of adjacent lung epithelial cells, namely malignant lung cancer cells in vitro and in murine xenograft models in vivo [86]. Increased expression of matrix metalloproteases (MMPs) as SASP were further found here in the RT-induced senescent lung fibroblasts, and a specific MMP inhibitor significantly restrained the growth of adjacent cancer cells [86].

A more detailed analysis of the secretory profile of senescent cells further revealed that secreted factors were highly correlated in senescent cells generated either by repeatedly passaging the cells (replicative exhaustion/senescence) or by exposing them to a relatively high dose (10 Gy) of ionizing radiation (SIPS) [87]. Inactivation of p53 in lung fibroblasts and the subsequently induced senescence by replication stress, radiation or the *RAS* oncogene revealed that the magnitude of SASP was significantly increased upon p53-deficiency, strongly suggesting that p53 is not required to initiate the SASP, but restrains development of an amplified SASP [87]. As decisive regulator of SASP, the transcription factor nuclear factor κB (NF-κB) was identified [9,88]. Proteome analysis of senescent chromatin revealed, that the NF-κB subunit p65 accumulated on chromatin of senescent fibroblasts [9]. In particular, the expression of many immune modulatory genes and secreted factors, including IL-6, IL-8, CXCL1, and ICAM1 were shown to depend on NF-κB expression, and more importantly, NF-κB suppression bypassed senescence and caused an immune escape by natural killer (NK) cells [9]. Among the identified SASP factors, IL-6 and IL-8 were the most likely candidates to increase the proliferation of adjacent premalignant and malignant epithelial cells, as well as stimulating epithelial-to-mesenchymal transition (EMT) and invasiveness in vitro and in vivo [87]. In addition, the persistence of senescent cells within lungs could contribute to a pro-inflammatory tissue environment and to the escape to immune surveillance [89]. Later on, the enhanced expression of the SASP factors CXCL12, HGF, MMPs and TGFβ in irradiated fibroblasts was reported to increase EMT and invasiveness of cancer cells, and enhanced expression of the SASP factors EGF, FGF-4, GM-CSF, IGF-1,2, IGFBP-2,4,6 induced chemoradioresistance in cancer cells [90]. The (in vitro) radiation-induced senescent fibroblasts herein were shown to share similar characteristics with activated fibroblasts and cancer-associated fibroblasts (CAF), the latter one arising predominately from normal fibroblasts that have been transformed by the tumor microenvironment [90]. Very recently, a comprehensive signalome analysis was performed using global gene expression profiling of irradiated fibroblasts (with different doses), replicative-aged fibroblasts and fibroblasts from old patients in order to gain insight into common signaling pathways affected by ionizing radiation and accelerated aging [91]. Herein, 12 h after irradiation for the doses 5 cGy and 2 Gy the suppression of replication and transcription (e.g., KLF4, VEGFA, ZNF691, DAB2IP), and enhancement of p38-MAPK, Wnt and VEGF pathway activities was observed. Later on (24 h after IR), a general suppression of gene expression was observed; in particular, a suppression of genes involved in DNA replication and G1/S transition, as well as a stimulation of hydrogen peroxide decomposition and glutathione synthesis (protection against ROS) [91]. Furthermore, the transcriptome of replicative-aged fibroblasts was more similar to the transcriptome of cells irradiated with higher doses [91]. Thus, this study identified important signaling pathways that are shared between senescence and irradiation processes that in turn could be used as a starting point to investigate potentially new targets of radioprotection [91].

Conclusively, there is clear evidence that radiation can induce senescence in lung fibroblasts, and that senescent lung fibroblasts affect adjacent (epithelial) cells. However, most of these investigations resulted from in vitro studies. Although senescence of fibroblasts seems to be an important feature in pulmonary fibrosis, the definitive contribution of radiation-induced senescence in lung fibroblasts and therefore their expected contribution to radiation-induced lung disease in vivo remains to be demonstrated. Investigating further strategies aiming to inhibit senescent fibroblast signaling by elimination, and/or normalization of senescence, paracrine signaling (SASP factor) blockade and extracellular matrix inhibition may hold a promising future for improving RT [90].

##### Senescence of Lung Endothelial Cells

Considerable in vitro and preclinical in vivo data support a deleterious impact of senescence on vascular endothelial cells finally resulting in the failure of the endothelium to perform its normal, physiologic functions [36,92,93]. It has been demonstrated that chronic clearance of senescent cells with senolytic drugs (e.g., dasatinib or quercetin), that selectively induce death of senescent cells or genetic clearance of p16-expressing endothelial cells, improves vascular phenotypes [45]. However, most of these studies focused on replicative senescence occurring during the processes of aging. Thus, the molecular mechanisms of endothelial senescence and associated vascular pathologies remain elusive, particularly with respect to radiation-induced lung injuries.

Nevertheless, a permanent cell-cycle arrest and a pro-inflammatory secretory phenotype could be induced by a variety of stimuli, including ionizing radiation, oxidative stress, and inflammation. In addition, prolonged TNFα exposure induced premature senescence in endothelial cells [38]. TNFα itself could induce the intracellular generation of ROS and is known to activate the inflammatory mediator NF-κB. Thus, chronically increased TNFα levels—potentially resulting from RT-affected neighboring cells—bear the potential to foster chronic inflammatory diseases by mediating endothelial senescence. Of note, anti-oxidant treatment (using N-acetyl cysteine) or NF-κB pathway inhibition (using the inhibitors plumericin and PHA-408) prevented TNFα-induced premature senescence of endothelial cells [38]. Radiation treatment itself was further shown to induce senescence in (cultured) quiescent microvascular endothelial cells in a dose- and time-dependent manner [24]. Besides the typical senescent cell cycle alterations, the persistent activation of the p53 pathway, mitochondrial dysfunction, and over-expression of the detoxification enzymes SOD2 (mitochondrial superoxide dismutase 2) and GPX1 (glutathione peroxidase 1) have been reported. Interestingly, the decreased induction of senescence in irradiated tissues of lungs upon treatment with the p53 inhibitor pifithrin-α or a superoxide dismutase mimetic (MnTBAP) revealed that IR-induced endothelial senescence involves different pathways (mitochondrial oxidative stress response or p53 activation), respectively; these observations suggest different potential strategies to reduce normal cell damage and modulate late radiation toxicity [24].

Although not derived from lung investigations, radiation-induced senescent glomerular endothelial cells acquired an altered SASP phenotype. Particularly, NF-kB activation and subsequent IL-6 signaling was characteristic for severe glomerular endothelial cell injury, as evidenced by thrombotic microangiopathy, collapsing glomeruli, and reduced endothelial cell numbers, respectively [37]. Generally, activation of pro-inflammatory cytokines (e.g., by tumor necrosis factor (TNF)-α or interleukin (IL)-6, led to activation of endothelial cells [94]. Mechanistically, endothelial cell activation results from activation of the NF-κB pathway. Genotoxic stress, including IR, with subsequent DBS-induced ATM activation led to the export of IKK-γ/NF-κB from the nucleus and thus NF-κB activation in the cytoplasm [25,95]. In addition, NF-κB as well as other redox-sensitive transcription factors [e.g., nuclear factor (erythroid-derived 2)-like 2 (NRF2) or activator protein 1 (AP-1)] were activated by (increased) ROS following γ-ray or X-ray treatments with a dose of 8–10 Gy [93]. Overall, different radiation qualities and doses of ionizing radiation induce different levels of oxidative stress, DSBs and/or shortening of telomeres, resulting in the activation of the hitherto quiescent endothelium and associated endothelial dysfunction—these processes are at least partially mediated by premature senescent endothelial cells [93,96,97]. Up to now, there was only one report which suggested that endothelial senescence contributed to radiation injury in lungs [22]. Herein, p16INK4a luciferase knock-in mice (a transgenic model imaging senescence in vivo by bioluminescence) were exposed to a single dose or fractionated radiation doses in a millimetric volume to investigate stress-induced senescence in the lungs following IR [22]. SA-β-gal-positive endothelial cells (besides alveolar epithelial cells), were mainly found in a ring area near the irradiated patch 12- and 15-months post-irradiation, which may contribute to the development of the severe injury in the small irradiated areas (corresponding to the ring around the lesion core).

Conclusively, the inhibition of oxidative stress and inflammation could be of great significance for controlling endothelial senescence, senescence-dependent structural and functional changes in the endothelium and associated vascular dysfunction. However, the role of radiation-induced endothelial senescence and its definitive contribution to radiation-induced lung disease in vivo remains to be demonstrated.

##### Senescence of Lung Epithelial Cells

Repetitive injury, especially to the pulmonary epithelium, is considered a central factor in the development of various lung diseases. Herein, the senescence of the respiratory epithelium either of the ciliated pseudostratified columnar epithelium, the cuboidal epithelium or the squamous epithelium in the alveolar ducts and alveoli is regarded as a central process for the initiation and progression of related lung diseases, particularly in pulmonary fibrosis and experimental lung fibrosis models [10,20,35]. Human lung tissues from lung fibrosis (IPF) patients were shown to harbor numerous senescent epithelial cells as revealed by prominent SA-β-gal and p16 staining [35]. IPF related epithelial senescence was closely associated with the SASP factors IL-1β, IL-6, IL-8 and TNF-α, which were already correlated with pulmonary fibrogenesis [98]. Of note, using an in vitro model of (bleomycin) stress-induced epithelial cell senescence, senescent lung epithelial cells-derived SASP factors were able to mediate the activation of pulmonary fibroblasts [35]. Therefore, the current hypothesis is that alveolar epithelial injury imposed on senescent epithelial cells leads to aberrant wound healing and the secretion of high levels of growth factors and chemokines that foster the activation of adjacent cells, including endothelial cells and fibroblasts, and the deposition of the ECM (Figure 1) [99]. Among the ‘activating’ epithelial-derived SAPS factors, increased levels of MMP12, SERPINE1, SPP1, and fibrotic mediator Wnt-inducible signaling protein (WISP) 1 were determined [100]. Moreover, pharmacological clearance of senescent lung epithelial cells by the induction of apoptosis in fibrotic alveolar (type II) epithelial cells or ex vivo three-dimensional lung tissue cultures (using dasatinib and quercetin) reduced SASP factors and extracellular matrix markers (e.g., collagen1a1, collagen5a3 and fibronectin) clearly indicating that senolytic drugs may be a viable therapeutic option for IPF [100].

In a preclinical model of radiation-induced pneumopathy, clearance of senescent cells with a senolytic drug (ABT-263) efficiently reduced senescent cells and reversed pulmonary fibrosis [26]. This, of course, would even limit the diminishing epithelial regenerative capacity, as well as associated SASP-mediated effects on adjacent lung cells as a central aspect in the development of lung injury. Therefore, targeting particularly senescent lung epithelial cells was suggested as a promising option for pulmonary fibrosis. Furthermore, radiation-induced senescence of lung epithelial cells was also closely connected to radiation-induced vascular dysfunction and associated extravasation of pre-metastatic immune and circulating tumor cells in a mouse model of radiation-induced pneumopathy [20,101]. Adoptive transfer of mesenchymal stem cells during the early phase after irradiation efficiently counteracted epithelial senescence (as well as vascular dysfunction) [21,101]. RT-induced senescence of bronchial-alveolar epithelial cells was further accompanied by the up-regulation of the SASP factor CCL2 [101]. Importantly, abrogation of certain aspects of the secretome of senescent lung cells, in particular signaling inhibition of the SASP factor CCL2, secreted predominantly by RT-induced senescent epithelial cells, limited inflammation as well as fibrosis progression [20]. This radioprotective action by addressing or modulating the SASP phenotype or senescent lung cells can have important implications in oncology, because higher doses of radiation might improve both local tumor control and survival. Moreover, treatment of thoracically irradiated mice with ABT-263 almost completely reversed pulmonary fibrosis, even when the initiation of ABT-263 treatment was delayed until fibrosis was established [26]. This means that unlike other known radiation protectants and mitigators, which were usually needed to be applied before or shortly after RT, senolytic drugs such as ABT-263 have the potential to be used as an effective, novel treatment of radiation-induced side complications such as inflammation and fibrosis, even after the lung injury develops into a progressive disease [70].

Although studies in animal models and patient samples show a complex response with multiple interactions between resident epithelial cells, fibroblasts and endothelial cells, and, in addition, infiltrating immune cells to radiation-induced diseased lung states, senescent epithelial cells and accompanied SASP contribute to the alteration and ‘activation’ of the lung microenvironment (Figure 1). The connection between senescence and infiltration of immune cells upon irradiation has been addressed elsewhere and is therefore not included here [102,103,104].

#### 3.2.2. Senescence of Lung Epithelial Cells: Cellular Stressors Other than RT

Senescence induction in lung epithelial cells was further observed to be important in chronic obstructive pulmonary disease (COPD), and may even play a role in the development of lung cancer. Although epithelial senescence was not induced by the stressor RT here, this section is included to emphasize again the central role of senescent lung epithelial cells in lung pathologies.

When cultured (human) epithelial cells were treated with serum, derived from blood of COPD patients, epithelial senescence was induced as revealed by increased SA-β-gal, histone γ-H2A.X, and p21 expression levels [27]. In addition, following this treatment, increased ROS levels as well as increased secretion of the SASP factors CXCL5, CXCL8/IL-8 and VEGF were determined in senescent epithelial cells. Increased ROS levels were already suggested to foster epithelial senescence, e.g., in lung epithelial cells exposed to nicotine [105]. The factors in the COPD serum need to be specified—in particular which noxa leads to the induction of senescence in lung epithelial cells. These ROS-derived SASP factors and/or the noxa itself might act on various adjacent cells in the lungs, e.g., endothelial cells and fibroblasts, as well as circulating immune cells and/or tumor cells [27]. The senescent phenotype of bronchial epithelial cells following COPD serum treatment was then shown to foster lung cancer cell proliferation and migration [27]. Epithelial senescence in COPD was already shown to be cigarette smoke-related. Mechanistically, cigarette smoke extract suppressed SIRT6, a histone deacetylase in cultured bronchial epithelial cells, and senescence induction was inhibited by SIRT6 overexpression [28]. SIRT6 overexpression further induced autophagy via attenuation of IGF-AKT-mTOR signaling. Autophagy inhibition in turn fostered the senescence-limiting effect of SIRT6 overexpression. Of note, SIRT6 expression levels were decreased in lung homogenates from COPD patients [28]. Loss of key anti-aging molecules may also be important in acceleration of aging and in particular in oxidative stress mediated senescence of lung epithelium. [48]. MicroRNA-34a, which is regulated by PI3K (phosphoinositide-3-kinase) and mTOR (mammalian target of rapamycin) signaling was shown to efficiently reduce the anti-aging factors sirtuin-1 and sirtuin-6. Its inhibition with an antagomir (also known as anti-miRs or blockmirs, chemically engineered oligonucleotides that prevent binding to a desired mRNA site) in turn resulted in restored sirtuin-1/6 levels, a reversed cell cycle arrest and thus reduced senescence and respective SASP in epithelial cells from the peripheral airways of patients with COPD [48]. Another key molecule involved in lung cellular senescence and thus in the inflammatory responses is miR-200b [29]. miR-200b was shown to be downregulated in the lungs of COPD model mice, while ZEB2 (Zinc finger E-box binding homeobox 2), a target gene of miR-200b, was increased. Treatment of cultured murine epithelial cells with cigarette smoke extract resulted in cellular senescence, while overexpression of miR-200b reduced cellular senescence and inflammatory responses. At the same time, the overexpression of miR-200b diminished the high ZEB2 protein expression that was induced by cigarette smoke extract [29]. Consequently, attenuating cellular senescence and the accompanied inflammatory responses by targeting miR-200b/ZEB2 in pulmonary emphysema might represent a novel therapeutic target for COPD and/or lung inflammation. Conformingly, the p53-induced senescence of club cells was further shown to promote chronic lung inflammation [39]. In a very elegant study, using mice whose club cells (the predominant bronchial epithelial cell type in mice) lack p53, the numbers of senescent bronchial epithelial cells were reduced upon chronic exposure to LPS (lipopolysaccharide). Those mice were protected from chronic lung inflammation and lung tissue destruction [39]. Treatment with a BCL-2 family inhibitor (ABT-737) further eliminated senescent cells in vivo, because senescent cells that were resistant to apoptosis, at least in part, upregulated the expression of members of the BCL-2. As a result, the senolytic treatment efficiently reduced chronic inflammation in the BAL fluid of mice subjected to chronic pro-inflammatory LPS exposure as well as in the lung parenchyma [39].

Conclusively, the reversal of senescence in terms of ‘rejuvenation’ in airway epithelial cells further highlights the use of drugs targeting cellular senescence (senotherapies) in chronic lung injuries.

### 3.3. Perspective: Biomarker Potential of SASP

Cellular senescence and a chronic SASP are known or suspected to be key drivers of many pathological hallmarks of lung injury, including age-related lung pathologies and lung diseases that are characterized by stress-induced premature senescence. The identification and establishment of senescence-associated biomarkers that drive aging and senescence-related diseases in specific tissues including the lung might improve detection and interventions in a time-dependent manner, so that disease progression might be prevented. Generally, during the senescence program, the transcriptome turned out to change dramatically from the time when it is first induced until an irreversible senescence phenotype is established [23]. In contrast to the short-term (acute) effects of senescent cells that are usually associated with positive effects because immune cells are recruited to remove senescent cells once they have executed their actions, the long-term (chronic) persistence of senescent cells is associated with disease due to the secretion of pro-inflammatory and pro-tumorigenic (SASP) factors [11]. In addition, the pro-longed cellular senescence critically affects the immune system with modulations of immune cell fate decision and immune responses in normal tissue homeostasis and in chronic inflammatory disorders [11,12,106,107]. Although the difficulty to track senescent cells in vivo and to perform disease-related quantifications of senescent cells in affected tissues including the lungs, non-invasive biomarkers of senescent cells could foster approaches in order to remove senescent cells and thereby improve the disease states.

In particular, SASP factors hold potential as plasma biomarkers for aging and diseases that are marked by the presence of senescent cells [108]. Herein, several serum metabolites, which were termed the ‘extracellular senescence metabolome (ESM)’, were identified in senescent fibroblasts either induced by proliferative exhaustion (aging) or by pre-matured senescence following irreparable DNA double strand breaks upon radiation treatment [23]. Asparagine, 2-hydroxybutyrate, 3-(4-hydroxyphenyl) lactate, 2-aminoadipate, alpha lipoate, two breakdown products of branch chain amino acid (BCAA) metabolism 3-hydroxyisobutyrate and isovalerate, Caprate (10.0), caprylate (8.0) and trans-4-hydroxyproline were identified as important radiation-related ESM components. Most of these components were implicated in ageing and mitochondrial dysfunction as well as in age-related diseases. Most consistently, extracellular citrate was suggested as specific ESM metabolite and thus as specific potential senescent biomarker [23]. Leukotrienes (LT) were further shown to be secreted from senescent cells regardless of the origin of the cells or the modality of senescence induction [43]. LT together with their eicosanoid family members prostaglandins (PGs), and COX2, the rate limiting enzyme in PG biosynthesis, were found to be highly upregulated during both normal and stress-induced fibroblast senescence. Thus, LT were identified as an additional SASP factor known to cause or exacerbate fibrosis [43]. In order to gain insight in the complexity of the SASP, a comprehensive proteomic database of soluble proteins and exosomal cargo SASP factors originating from multiple senescence inducers and cell types, the so called SASP Atlas (www.SASPAtlas.com) was presented [108]. Proteins secreted by senescent and quiescent/control primary human lung fibroblasts and (renal) epithelial cells were analyzed by mass spectrometry following senescence induction by X-irradiation (IR), inducible RAS overexpression (RAS), or atazanavir (a protease inhibitor used in HIV treatment) stimulation. This unbiased proteomic profiling identified numerous secreted proteins per senescence inducer. Among these secreted factors were well-known SASP factors, e.g., CXCLs, high mobility group box 1 protein (HMGB1), insulin-like growth factor binding proteins (IGFBPs), matrix metalloproteinases (MMPs), laminin subunit beta-1 (LAMB1), and tissue inhibitors of metallopeptidase (TIMPs) [108]. K-means clustering analysis further identified ‘core SASP’ proteins, which were represented in the senescent secretome of all inducers, namely the chemokine C-X-C motif ligand 1 (CXCL1), MMP1, and stanniocalcin 1 (STC1) as well as plasminogen activator inhibitor 1 (SERPINE1) and GDF15 (growth/differentiation factor 15). Thus, these proteins may serve as surrogate SASP markers. The epithelial-related SASP could be categorized into three general terms/pathways: protein turnover and secretion, primary metabolism, and cellular detoxification. Furthermore, factors like IGFBPs 4/7, TIMPs 1 and 2, CXCL1, and most serine protease inhibitors (SERPINs) were alternatively changed upon secretion of epithelial and fibroblast cells [108]. Although no IR-specific SASP signature could be stated, a oncogene-induced senescence signature was established consisting of the five proteins—CXCL5, MMP9, MMP3, Cystatin-S (CST4), and C-C motif chemokine 3 (CCL3) [108]. In this respect, it has to be noted that many of these SASP factors are also present in premalignant lesions and certain cancers demonstrating the same patterns as senescent cells [109]. It was even reported that senescent cells can escape from its cell cycle arrest to form cancers [110]. Thus, the final validity of the biomarker potential of certain SASP factors still needs to be confirmed in vivo.

Conclusively, this SASP database is a new, valuable and promising approach to identifying and comparing candidate biomarkers of aging and diseases driven by senescent cells like lung injuries. Whether different senescence inducers produce similar or distinct SASPs is, at present, poorly characterized. Thus, a comprehensive characterization of SASP components is critical to understand how senescent responses can drive diverse pathological phenotypes in vivo.

## 4. Conclusions

Depending on the extent of the damage and the physiological relationships, DNA damage switches the DNA repair machinery on and triggers apoptosis or senescence. Replicative senescence is triggered by telomere shortening in which normal, non-malignant cells stop dividing once the telomeres have reached the critical minimum, and thus DNA damage persists. Prolonged DNA damage and subsequent senescence can also result from different triggers: chromatin alterations, activation of certain oncogenes, and genotoxic therapeutic drugs, such as some chemotherapies or therapies for HIV treatment, and in particular from radiation treatment. All the presented reports suggest that cellular senescence contributes importantly to the pathogenesis of different lung diseases and specific targeting of senescent cells represents a potential approach for the treatment of these lung disorders. Different types of lung cells, namely bronchial and alveolar epithelial cells, fibroblasts, and endothelial cells, have been shown to undergo senescence in diseased lungs. Due to the overwhelming evidence showing that pulmonary epithelial cells undergo senescence in response to various stimuli, we suggest here a central role of epithelial senescence (Figure 1). The cellular stress-induced senescence of lung epithelial cells, and the accompanied expression of diverse secreted proteins termed SASP, regulate key processes that facilitate further senescence either in an autocrine loop or via paracrine signaling to adjacent lung cells, proliferation, activation, angiogenesis, inflammation and tissue remodeling finally driving lung disease progression. Thus, the potential of future therapeutics could be to address the induction and cell type specific cellular senescence, which not only determines the susceptibility to certain lung diseases, but also bears the potential to disrupt this process in terms of new treatment options. Radiation-induced lung toxicity for example, which is at least partially mediated by radiation induced-lung epithelial senescence, often precludes the application of curative radiation doses. Senescence targeting therapies might therefore not only rationally prevent or minimize late effects and maximize a survivor’s quality of life, it even bears the potential to reverse established radiation-induced late adverse effects, namely pulmonary fibrosis.

## Figures and Tables

**Figure 1 ijms-21-03279-f001:**
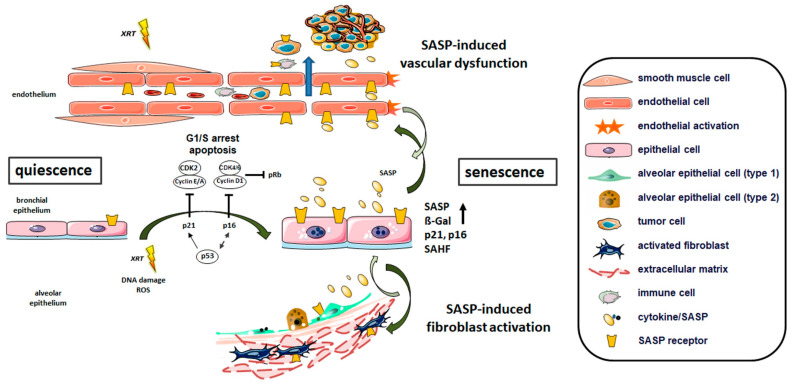
Cellular senescence orchestrating lung injury. Upon thoracic irradiations (XRT, as shown by the yellow flash), normal lung tissue reacts with complex and multiple interactions between resident cells: bronchial and alveolar epithelial cells, fibroblasts, and endothelial cells. Senescence of the respiratory epithelium is considered here as a central process for the initiation and progression of related lung diseases, particularly the development of pneumonitis and fibrosis (radiation-induced pneumopathy). Persistent or irreparable DNA damage following XRT in lung epithelial cells in turn can induce an irreversible cell cycle arrest that lead to apoptosis or to the establishment of cellular senescence. Mechanistically, senescence is induced and maintained by signaling cascades that activate p53/p21 and/or p16/Rb tumor suppressor pathways that inhibit epithelial cell cycle progression. Accordingly, increased p16 and p21 transcript levels can be observed. Senescent epithelial cells further bear characteristic morphological features: a larger and flat-like cell morphology, an increase in senescence-associated β-galactosidase (SA-β-gal) activity, and, in some cell types, a discernible change in chromatin organization known as senescence-associated heterochromatin foci (SAHF) that are marked by foci of histone H3 lysine 9 trimethylation. In addition, the proinflammatory and pro-oxidative senescence-associated secretory phenotype (SASP) of these cells, could reinforce the senescence arrest, alter the microenvironment and thus impair the function of neighboring cells in a paracrine manner. Radiation-induced epithelial senescence leads to increased SASP factor production. Very close by, the hitherto quiescent healthy endothelium that usually provides an efficient barrier to liquids or cell extravasation becomes activated and/or bears an “angiogenic” phenotype (acute effect) in response to certain SASP factors. Increased endothelial permeability associated with increased leakage of blood stream components into the lung interstitium then fosters inflammation and/or metastasis formation. Normal fibroblasts will also be activated by SASP factors potentially resulting in a phenotypic change into pro-fibrotic myofibroblasts and/or cancer-associated fibroblasts that foster tissue remodeling by extracellular matrix deposition and thus fibrosis progression. The cross-talk between senescent epithelial cells and adjacent endothelial cells and fibroblasts is bi-directional. Radiation-induced senescence of lung endothelial cells and fibroblasts could even contribute to lung injury (not depicted). Clearance of senescent lung epithelial cells or abrogation of certain aspects of their secretome may represent a novel target for the treatment of inflammatory and/or fibrotic lung disorders.

**Table 1 ijms-21-03279-t001:** Summary of methods for the detection of senescent cells.

Target	Marker		Method of Detection
Lysosomes	SA-β-gal		Histochemical detection of β-galactosidase activity at pH 6 [9,20,22,23,24,25,26,27,28,29]
			Fluorogenic probes (e.g., C12FDG) [30,31]
			Near-infrared molecular probe (in vivo and in vitro) [32]
			Two-photon fluorescent probe (in vivo and in vitro) [33]
	Lipofuscin		Lysosomal aggregates stained with Sudan Black B [34]
Cell cycle inhibitors	p16^INK4a^, p21^Cip/Waf1^, p15^INK4b^, p27		Western blot [9,18,19,20,23,24,29,35]
			RT-PCR [20,35,36,37]
			Immunofluorescence [35,38]
			Immunohistochemistry [29,39]
Cell proliferation	Ki-67 (absence)		Western Blot [40]
			RT-PCR [40]
			Immunofluorescence [38]
	BrdU incorporation (absence)		Immunofluorescence [18]
	Telomere shortening		FISH [41,42]
SASP factors	Cytokines (e.g., IL-6, TNFα) Chemokines (e.g., IL-8, MIPs, CCLs) Proteases (e.g., MMPs) Candidates: TGFβ, GM-CSF, PAI-1, IGF-1		Immunofluorescence [19,20]
			RT-PCR [9,20,25,26,37,38,43]
			Western Blot [9,19,20]
Tumor suppressors	pPTEN, p53, hypo-phosphorylated Rb, FOXO4		Western blot [9,18,24]
			RT-PCR [36]
			Immunofluorescence [6]
Chromatin organization	SAHF	NFκB p65 subunit	Immunofluorescence [9]
			Western Blot [9]
			RT-PCR [9]
		reorganization of DNA structure by DAPI, antibodies against facultative heterochromatin	Immunofluorescence [44]
DNA damage marker	γH2AX		Western blot [45]
			Immunofluorescence [27,28]

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
