# Peer review of "Cellular Senescence in the Lung: The Central Role of Senescent Epithelial Cells"

_ijms, 2020, doi:10.3390/ijms21093279_

Round 1

Reviewer 1 Report

I read with a lot of interest the review article “Cellular senescence in the lung: the central role of senescent epithelial cells”. The review is well-organized. The review clearly described in the first chapter replicative and stress-induced senescence. Then, role of fibroblast, endothelial and epithelial senescence after irradiation is explained. The authors should insist of the difficulty to track senescent cells in vivo quantification of senescent cells. In fact, a lot of data of the review are obtained with cell culture models.

The 3 chapters on the senescence of the 3 of the major cell types in lung (fibroblast, endothelial cell and epithelial cells). Because of its role in fibrosis and its recruitment by proinflammatory SASP, A statement on the link between Immune cells activation and recruitment by SASP factor will be helpful. Furthermore, it is not clear why senescence of epithelial cells is more important in Lung radiopathology than endothelial or fibroblast senescence.

Finally, the discussion of the mode of radiotherapy hypofractionation or Flash should be either removed or at least localized in section 2.1. However, I don’t really understand the interest of the sentence on the Flash radiotherapy.

In conclusion, this review on lung senescence after radiotherapy is well-written and useful for the radiobiology/radiotherapy community. The need of overview of the immune response by SASP and inflammation should give a whole overview of the topic.

Author Response

Point to point response to the Reviewers’ comments

Reviewers’ Comments:

Reviewer 1

I read with a lot of interest the review article “Cellular senescence in the lung: the central role of senescent epithelial cells”. The review is well-organized. The review clearly described in the first chapter replicative and stress-induced senescence. Then, role of fibroblast, endothelial and epithelial senescence after irradiation is explained. The authors should insist of the difficulty to track senescent cells in vivo quantification of senescent cells. In fact, a lot of data of the review are obtained with cell culture models.

Answer

First of all, we would like to thank for the positive evaluation of our manuscript enabling us the re-submission of our revised manuscript. According to the reviewer’s suggestion we addressed now the critical points. We included now a paragraph about the difficulty to track senescent cells in vivo (within the introduction and at the beginning of section 2.7) and included a table summarizing the different markers/methods for the detection of senescent cells (new Table 1).

Changes were emphasized within the manuscript using yellow color.

Comment 2

The 3 chapters on the senescence of the 3 of the major cell types in lung (fibroblast, endothelial cell and epithelial cells). Because of its role in fibrosis and its recruitment by proinflammatory SASP, A statement on the link between Immune cells activation and recruitment by SASP factor will be helpful. Furthermore, it is not clear why senescence of epithelial cells is more important in Lung radiopathology than endothelial or fibroblast senescence.

Answer 2

We agree with the reviewer that this is an important issue. According to the reviewer’s suggestion we included a statement on the link between immune cells activation and recruitment by SASP factor and included respective references here (in section 2.7). However, a detailed summary about the different subsets of immune cells is not provided in this review. We included that limitation at the end of the section 2.5. We further re-arranged the information in the different sections (see answer to comment 3), and further strengthen why senescence of epithelial cells play a central role in lung diseases (e.g. section 2.5 first paragraph and last paragraph).

Comment 3

Finally, the discussion of the mode of radiotherapy hypofractionation or Flash should be either removed or at least localized in section 2.1. However, I don’t really understand the interest of the sentence on the Flash radiotherapy.

Answer 3

According to the reviewers suggestion we localized this information in initial section 2.1 (which is now 2.2). We further re-arranged the information in the different sections (according to the suggestions of reviewer 2, comment 7) to improve precision and clarity and separated the recent findings about stress (radiation)-induced senescence and replicative senescence.

Comment 4

In conclusion, this review on lung senescence after radiotherapy is well-written and useful for the radiobiology/radiotherapy community. The need of overview of the immune response by SASP and inflammation should give a whole overview of the topic.

Answer 4

We refer to the answer of comment 2.

Reviewer 2 Report

The review by Hansel et al aims at describing the role of senescence in different lung pathologies mainly of fibrotic nature. Stating the authors, a « focus on radiation-induced senescence » is expected. Authors also distinguish three cell type categories : epithelial, endothelial and fibroblasts, which could draw interest in the biological mechanisms. Unfortunately, the manuscript lacks precision and clarity, and does meet the presented goals. The focus on ionizing radiation is scattered through different sections, including one named « radiation… ». Different biological pathways, pathologies and types of senescence are mixed together. Overall, the review has inherent problems within its structure, is not didactic, and the writing of scientific facts is often unprecise leading to approximation, if not intuitively wrong. It is extremely confusing and not clearly understandable for naive readers. Some statements are somewhat unclear and confusing.

Below is a non-exhaustive list of the recurring problems :

1- Authors state line 32-34 that senescence was discovered by Leonard Hayflick because he showed programmed cell death after a limited number of divisions. Cell death is not the definition of senescence.

2-Lines 39-42, authors present the p53-p21-p16 axis as « irreversible cell cycle arrest ». It is not the case in every biological situation, and can lead to misunderstanding in the context of alternative cellular processes.

3- Unlike what is stated line 44, fibroblasts are not large and flat.

4- The markers presented in lines 49-52 cannot be used to distinguish senescence from any other resembling condition.

5- A table recapitulating the different markers, their use etc.. would have been informative at this place.

6- It seems contradictory to say that « cellular senescence is now considered as an important driving force in chronic lung pathologies » line 70, and « little is known how it impacts on lung disease phenotype » line 74.

7- There are two major types of senescence, namely stress-induced senescence and replicative senescence. These differ drastically in their originating biological mechanisms. This should be readily explained at the begininng of the manuscript. The comment line 91 is unsufficient to prepare readers for good understanding. There is continuous mixing of replicative and stress-induced senescence in the different sections.

8- The comment line 83 regarding age is confusing. Premature senescence is not strictly an age-dependent mechanism, which is why it is called premature.

9- In 2.2, an introduction explaining the physiological and pathological roles of fibroblasts in the lung would have been better that just a comment further line 151. This paragraph is a list of statements that are not well articulated and from which it is hard to depict the essential information.

10- « Sedentary aging » line 177 is not appropriate in section 2/stress-induced senescence.

11- Antecubital veins and aging line 182 is not relevant as well in this section.

12- « were further cultured to senescence » line 186 pertains to replicative senescence. Again, it is not related to stress-induced senescence.

13- Comment line 219 refers to a publication on aging, still not relevant to section 2.

14- Describing the different cell populations which can be affected by –and affect-  senescence is of interest. However, the contribution of each cell type to lung pathology is largely unadressed. For instance, authors should describe how fibroblasts induce fibrosis and how senescence regulates this mechanism.

15- Line 242-259 : mechanisms involving telomerase activity relate to replicative senescence, not stress-induced.

16- Line 260-270 : evidence of the role of senescence in lung pathology should be presented first. Moreover, this paragraph deals mostly with fibroblasts, which are the scope of the section 2.2, not 2.4.

17- Line 271-279 : should be presented before also.

18- Line 280-317 deals with radiotherapy and should be presented in section 2.1 : « radiation-induced cellular senescence in the lungs ».

19- Section 2.5 : it must be noted that most of the markers presented here are also frequently found in other pathological situations such as cancer. This should be discussed.

20- Figure 1 : the figure presents unidirectional sasp signaling from epithelial cells to adjacent vessels and stroma. Reciprocal signaling should be presented to make the figure complete. Moreover, it is not clear why there is a red cross on apoptosis and how cells transit from quiescence to senescence on the illustration.

21- The figure legend, line 420-463 is not a legend. It is too long, and deals with items that are not presented in the figure.

22- There are numerous syntax and grammatical errors. Some sentences are extensively long to be fairly understood.

Author Response

Point to point response to the Reviewers’ comments

Reviewers’ Comments:

Reviewer 2

The review by Hansel et al aims at describing the role of senescence in different lung pathologies mainly of fibrotic nature. Stating the authors, a « focus on radiation-induced senescence » is expected. Authors also distinguish three cell type categories : epithelial, endothelial and fibroblasts, which could draw interest in the biological mechanisms. Unfortunately, the manuscript lacks precision and clarity, and does meet the presented goals. The focus on ionizing radiation is scattered through different sections, including one named « radiation… ». Different biological pathways, pathologies and types of senescence are mixed together. Overall, the review has inherent problems within its structure, is not didactic, and the writing of scientific facts is often unprecise leading to approximation, if not intuitively wrong. It is extremely confusing and not clearly understandable for naive readers. Some statements are somewhat unclear and confusing.

Answer

First of all, we would like to thank for the critical evaluation of our manuscript. According to the reviewer’s suggestion, we addressed and corrected the critical points. Changes were emphasized within the manuscript using yellow color.

Below is a non-exhaustive list of the recurring problems :

1- Authors state line 32-34 that senescence was discovered by Leonard Hayflick because he showed programmed cell death after a limited number of divisions. Cell death is not the definition of senescence.

Answer 1

We apologize for that unclear description and corrected the respective sentence.

2-Lines 39-42, authors present the p53-p21-p16 axis as « irreversible cell cycle arrest ». It is not the case in every biological situation, and can lead to misunderstanding in the context of alternative cellular processes.

Answer 2

We absolutely agree with the critic of the reviewer and improved the respective section (now lines 40-49).

3- Unlike what is stated line 44, fibroblasts are not large and flat.

Answer 3

This has been corrected.

4- The markers presented in lines 49-52 cannot be used to distinguish senescence from any other resembling condition.

5- A table recapitulating the different markers, their use etc.. would have been informative at this place.

Answer 4/5

We absolutely agree that this is an important issue. We improved the paragraph (now line 59 and onwards) concerning the combination of senescence markers and the difficulty to track senescent cells in vivo. According to the reviewers suggestion we further included now a table summarizing the different markers/methods for the detection of senescent cells (new Table 1).

6- It seems contradictory to say that « cellular senescence is now considered as an important driving force in chronic lung pathologies » line 70, and « little is known how it impacts on lung disease phenotype » line 74.

Answer 6

This has been corrected.

7- There are two major types of senescence, namely stress-induced senescence and replicative senescence. These differ drastically in their originating biological mechanisms. This should be readily explained at the begininng of the manuscript. The comment line 91 is unsufficient to prepare readers for good understanding. There is continuous mixing of replicative and stress-induced senescence in the different sections.

Answer 7

We thank the reviewer for this helpful comment. According to the reviewers suggestion we included a new section (2. Replicative senescence versus stress-induced senescence) at the beginning of the manuscript. We further re-arranged the information in the different sections to improve precision and clarity and separated the recent findings about stress (radiation)-induced senescence and replicative senescence. The paragraph 2.1 (Cellular senescence in adult lungs) highlights now the importance of senescence (in general) in lungs, and 2.2 (Radiation-induced cellular senescence in lungs) proceeds with stress-induced senescence, particularly radiation-induced senescence.

8- The comment line 83 regarding age is confusing. Premature senescence is not strictly an age-dependent mechanism, which is why it is called premature.

Answer 8

This has been corrected. As mentioned in the answer to the comment before

9- In 2.2, an introduction explaining the physiological and pathological roles of fibroblasts in the lung would have been better that just a comment further line 151. This paragraph is a list of statements that are not well articulated and from which it is hard to depict the essential information.

Answer 9

We absolutely agree that this is an important issue. We completely revised that section (which is now 2.3) and as suggested by the reviewer we included a paragraph about the physiological and pathological roles of fibroblasts in the lung at the beginning of that section.

10- « Sedentary aging » line 177 is not appropriate in section 2/stress-induced senescence.

11- Antecubital veins and aging line 182 is not relevant as well in this section.

12- « were further cultured to senescence » line 186 pertains to replicative senescence. Again, it is not related to stress-induced senescence.

13- Comment line 219 refers to a publication on aging, still not relevant to section 2.

Answer 10-13

We refer to the answer of comment 7. The findings in the different sections have been separated concerning stress (radiation)-induced and replicative senescence.

14- Describing the different cell populations which can be affected by –and affect-  senescence is of interest. However, the contribution of each cell type to lung pathology is largely unadressed. For instance, authors should describe how fibroblasts induce fibrosis and how senescence regulates this mechanism.

Answer 14

We refer to the answer of comment 7. This information has been introduced now in the respective sections (now section 2.1, 2.2 and 2.3, beginning).

15- Line 242-259 : mechanisms involving telomerase activity relate to replicative senescence, not stress-induced.

Answer 15

This has been corrected. We refer to the answer of comment 7 and of comment 10-13.

16- Line 260-270 : evidence of the role of senescence in lung pathology should be presented first. Moreover, this paragraph deals mostly with fibroblasts, which are the scope of the section 2.2, not 2.4.

17- Line 271-279 : should be presented before also.

18- Line 280-317 deals with radiotherapy and should be presented in section 2.1 : « radiation-induced cellular senescence in the lungs ».

Answer 16-18

This has been corrected/ re-arranged. We refer to the answer of comment 7 and of comment 10-13.

19- Section 2.5 : it must be noted that most of the markers presented here are also frequently found in other pathological situations such as cancer. This should be discussed.

Answer 19

Again, we would like to thank the reviewer for this important point. We included that limitation in the respective section (which is now section 2.7).

20- Figure 1 : the figure presents unidirectional sasp signaling from epithelial cells to adjacent vessels and stroma. Reciprocal signaling should be presented to make the figure complete. Moreover, it is not clear why there is a red cross on apoptosis and how cells transit from quiescence to senescence on the illustration.

21- The figure legend, line 420-463 is not a legend. It is too long, and deals with items that are not presented in the figure.

Answer 20/21

According to the reviewers suggestion we included the bi-directional crosstalk between epithelial cells and adjacent cells in the figure as well as in the figure legend. We further strengthen why senescence of epithelial cells play a central role in lung diseases (now section 2.5 and 2.6), and improved the figure legend.

22- There are numerous syntax and grammatical errors. Some sentences are extensively long to be fairly understood.

Answer 22

We apologize for the mistakes in writing. We critically checked and corrected the syntax and grammatical errors throughout the manuscript and improved/ shortened the sentences.

Round 2

Reviewer 2 Report

The review by Hansel et al deals with the role of cellular senescence in lung pathologies, with a focus on irradiation. Although the topic was of interest, the initial manuscript suffered from confusing descriptions and structure. To be honnest with the authors, my initial decision was to reject the manuscript because, from my opinion, it required to almost rewrite entirely the review.

In this second version, I acknowledge that authors have performed significant work. The changes made clearly improve the manuscript, both in term of scientific accuracy than in term of understanding. In particular, the description of the role of the different cell types and the contribution of their senescence to lung pathology, gives a better view of why the authors indicate "central role of senescent epithelial cells" in the title. For naive readers, the table 1 and section 2 (replicative vs stress-induced) represent valuable ressources.

Based on this, I am willing to accept to change the status of the manuscript to minor revisions. It is my call to trust that the authors will indeed perform new changes to further improve their manuscript. A list of the principal items is given below:

1- There remains apparent confusion in the structure. At this point, it seems that it is more a problem drawn by the titles of the different sections rather than by the content of these sections. Please make sure the titles describe accurately the content below them. For instance, section 2 is called "replicative senescence vs stress-induced senescence". This would be appropriate for the paragraph below it, but not for the entire section 2. Remember that sections 2.1, 2.2 etc... are also part of section 2. Also, section 2.1 deals mostly with age-related senescence. This might be indicated in the corresponding title. Section 2.2 seems to relate to pathophysiology, etc...

2- The legend of figure 1 is still long. It seems that there are many sentences that relate to general pathophysiology and could be for example moved to the beginning of section 2.Also, the description fo the different epithelial types could be moved to section 2.5.

3- Table 1 needs better consistency. For instance, if authors indicate "absence" for Ki67, they must do the same for brdu incorporation. Western-blot can also be used for SASP factors, etc...

4- There remains few writing mistakes (grammar and typos) that should be easily corrected by the authors and during editing by the publisher.

Author Response

Point to point response to the Reviewers’ comments

Reviewers’ Comments:

Reviewer 2

The review by Hansel et al deals with the role of cellular senescence in lung pathologies, with a focus on irradiation. Although the topic was of interest, the initial manuscript suffered from confusing descriptions and structure. To be honest with the authors, my initial decision was to reject the manuscript because, from my opinion, it required to almost rewrite entirely the review.

In this second version, I acknowledge that authors have performed significant work. The changes made clearly improve the manuscript, both in term of scientific accuracy than in term of understanding. In particular, the description of the role of the different cell types and the contribution of their senescence to lung pathology, gives a better view of why the authors indicate "central role of senescent epithelial cells" in the title. For naive readers, the table 1 and section 2 (replicative vs stress-induced) represent valuable ressources.

Based on this, I am willing to accept to change the status of the manuscript to minor revisions. It is my call to trust that the authors will indeed perform new changes to further improve their manuscript. A list of the principal items is given below:

Comment 1

1- There remains apparent confusion in the structure. At this point, it seems that it is more a problem drawn by the titles of the different sections rather than by the content of these sections. Please make sure the titles describe accurately the content below them. For instance, section 2 is called "replicative senescence vs stress-induced senescence". This would be appropriate for the paragraph below it, but not for the entire section 2. Remember that sections 2.1, 2.2 etc... are also part of section 2. Also, section 2.1 deals mostly with age-related senescence. This might be indicated in the corresponding title. Section 2.2 seems to relate to pathophysiology, etc...

Answer1

First of all, we would like to thank the reviewer now for the positive evaluation of our manuscript enabling us the re-submission of our revised manuscript. According to the reviewer’s suggestion we (again) addressed now the critical points to further improve our manuscript. Changes were emphasized within the manuscript using green color.

According to the reviewer’s suggestion we re-numbered the different paragraphs to describe more accurately the content below, and to clarify better which sections belong together.

Comment 2

2- The legend of figure 1 is still long. It seems that there are many sentences that relate to general pathophysiology and could be for example moved to the beginning of section 2. Also, the description of the different epithelial types could be moved to section 2.5.

Answer 2

We shortened the legend of Figure 1. As suggested by the reviewer we moved the indicated statements to the respective sections.

Comment 3

3- Table 1 needs better consistency. For instance, if authors indicate "absence" for Ki67, they must do the same for brdu incorporation. Western-blot can also be used for SASP factors, etc...

Answer 3

As suggested by the reviewer we improved Table 1.

Comment 4

4- There remains few writing mistakes (grammar and typos) that should be easily corrected by the authors and during editing by the publisher.

Answer 4

We again apologize for the mistakes in writing. We critically checked and corrected all the errors.